# Sleep Monitoring during Acute Stroke Rehabilitation: Toward Automated Measurement Using Multimodal Wireless Sensors

**DOI:** 10.3390/s22166190

**Published:** 2022-08-18

**Authors:** Pin-Wei Chen, Megan K. O’Brien, Adam P. Horin, Lori L. McGee Koch, Jong Yoon Lee, Shuai Xu, Phyllis C. Zee, Vineet M. Arora, Arun Jayaraman

**Affiliations:** 1Max Nader Lab for Rehabilitation Technologies and Outcomes Research, Shirley Ryan Ability Lab, Chicago, IL 60611, USA; 2Department of Physical Medicine and Rehabilitation, Northwestern University, Chicago, IL 60611, USA; 3Sibel Health Inc., Niles, IL 60714, USA; 4Querrey Simpson Institute for Bioelectronics, Northwestern University, Evanston, IL 60208, USA; 5Center for Circadian and Sleep Medicine, Department of Neurology, Northwestern University, Chicago, IL 60611, USA; 6Department of Medicine, University of Chicago Medicine, Chicago, IL 60637, USA

**Keywords:** machine learning, stroke, sleep, rehabilitation, wearable sensors, health outcome

## Abstract

Sleep plays a critical role in stroke recovery. However, there are limited practices to measure sleep for individuals with stroke, thus inhibiting our ability to identify and treat poor sleep quality. Wireless, body-worn sensors offer a solution for continuous sleep monitoring. In this study, we explored the feasibility of (1) collecting overnight biophysical data from patients with subacute stroke using a simple sensor system and (2) constructing machine-learned algorithms to detect sleep stages. Ten individuals with stroke in an inpatient rehabilitation hospital wore two wireless sensors during a single night of sleep. Polysomnography served as ground truth to classify different sleep stages. A population model, trained on data from multiple patients and tested on data from a separate patient, performed poorly for this limited sample. Personal models trained on data from one patient and tested on separate data from the same patient demonstrated markedly improved performance over population models and research-grade wearable devices to detect sleep/wake. Ultimately, the heterogeneity of biophysical signals after stroke may present a challenge in building generalizable population models. Personal models offer a provisional method to capture high-resolution sleep metrics from simple wearable sensors by leveraging a single night of polysomnography data.

## 1. Introduction

It is increasingly clear that there is an important connection between stroke and sleep. A recent observational study of over 80,000 people found that individuals with insomnia had a 54% increased risk of stroke in the ensuing 4 years compared to age-matched non-insomniacs [1]. After stroke, an estimated 27% of patients report hypersomnia and excessive daytime sleepiness [2], while an estimated 57% of patients report insomnia [3]. Poor sleep has a detrimental impact on recovery, with delayed neuroplasticity and motor learning [4]. Increases in the degree of insomnia are negatively correlated with Barthel index, which measures positive changes in physical function during daily activities [5]. Alternatively, early animal studies have also shown that an increase in slow wave sleep after a stroke (i.e., using optogenetic stimulation) induces positive sleep-dependent plasticity resulting in better functional recovery [6]. Together, these studies emphasize that poor sleep can impede recovery after stroke, and that improving sleep could be one strategy to lead to better patient outcomes.

Before we can provide meaningful interventions to improve sleep and neural recovery for patients with stroke, it is essential to accurately measure sleep and identify deficits. Currently, there are limited practices in place to evaluate sleep quality in a hospital setting. The most common approach is ordering a sleep log, in which the care team manually checks on the patients periodically throughout the night and notes whether they are awake or asleep. This practice has low sensitivity to measure sleep *quantity* and provides no indicator of *quality*, especially in an acute stroke rehabilitation setting in which a certain sleep stage (e.g., slow wave sleep) is important for recovery, and sleep time is fixed by care schedule. At the other extreme, polysomnography (PSG), which consists of electroencephalogram (EEG), photoplethysmography (PPG), electrocardiogram (ECG), capnography, and/or respiration measurements, is the current gold standard technique to evaluate sleep architecture in clinical or research settings [7]. However, PSG equipment for a widespread, long-term implementation is impractical in an inpatient rehabilitation facility (IRF) due to the high system cost and high workload to collect and analyze. Furthermore, patients often report discomfort wearing numerous wired devices, which are bulky and can disrupt their sleep.

Wireless, wearable sensors could address these limitations. As engineering advances, these devices are becoming more flexible, lightweight, and cost effective. These sensors also have the capability to collect heart rates, body temperatures, or oxygen saturation that can be biomarkers for quantifying the different sleep stages through the modulation of the autonomic nervous system (ANS). ANS activities are often coupled with neuronal activities at the central nervous system (CNS) [8,9,10,11]. This CNS-ANS coupling network directly affect cardiovascular responses during sleep. Today, many commercial activity trackers and smartwatches provide sleep quality measures and sleep staging, though the accuracy of these devices for individuals with stroke is often unclear. Sensors that provide access to raw biophysical signals, when paired with advanced machine learning algorithms, offer the ability to construct sleep detection algorithms from scratch and optimize their performance for specific use cases. We have previously demonstrated that an algorithm trained from a set of low-profile sensors measuring motion, heart activity, and skin temperature was promising to detect different sleep stages (i.e., light sleep, deep sleep, or rapid eye movement (REM) sleep) in healthy individuals [12]. To our knowledge, no algorithms have been constructed or validated for patients with stroke, whose ANS signals can be very different from healthy individuals due to their neurological injury [13] or medications used during treatment (e.g., beta blockers to manage heart rhythms).

In this study, we explored the feasibility of evaluating overnight sleep for patients with subacute stroke using multimodal wearable sensors. Similar to our previous work with healthy individuals [12], we obtained overnight sleep data from a preliminary sample of patients using PSG and wearable sensors. We tested two types of supervised machine learning approaches: population models (trained on data from a subset of patients and tested on data from a left-out patient) and personal models (trained on a subset of data from one patient and tested on left-out data from the same patient). Insights from this preliminary work can inform future studies of sleep monitoring for patients with stroke. High-resolution, objective sleep monitoring with wearable sensors would enable us to unobtrusively identify individuals at risk for poor, non-restorative sleep across care settings. In turn, this will empower clinicians and researchers to develop personalized interventions for improving sleep—and thus enhancing neural recovery during acute rehabilitation—for these individuals.

## 2. Materials and Methods

### 2.1. Participants

Ten individuals with stroke (5F/5M; age 58.1 ± 12.1 years) were recruited from the inpatient unit of the Shirley Ryan AbilityLab, a rehabilitation facility in Chicago, IL (USA), for a single night of sleep monitoring with PSG and wearable sensors. The Institutional Review Board Office at the Northwestern University (STU00206700) approved the protocol, and all patients provided informed, written consent prior to participation. Participants were required to be at least 18 years of age, understand spoken English at a sixth-grade level or higher, and have a primary diagnosis of stroke. Patients with diagnosed sleep disorders (e.g., obstructive or central sleep apnea) were excluded from the study at this preliminary stage. Table 1 summarizes the demographics and clinical characteristics of the 10 participants. Table A1 (Appendix A) provides additional attributes about stroke for each participant.

### 2.2. Equipment

#### 2.2.1. Polysomnography (PSG)

Electroencephalographic (EEG), electrooculographic (EOG), and submental electromyographic (EMG) signals, three-lead ECG, and a respiratory belt were recorded on a portable PSG system (Brain Vision; Morrisville, NC, USA). Ten EEG channels were applied following the international 10–20 system (ROC, LOC, C3, C4, F3, F4, P3, P4, O1, and O2).

#### 2.2.2. Wearable Sensors

ANNE^TM^ One (Sibel Health; Niles, IL, USA) is an FDA-cleared, clinical-grade sensor system with two soft, flexible devices: one adhered to the chest using an adhesive sticker to measure triaxial acceleration as well as ECG, heart rate, respiratory rate, and proximal skin temperature, and one placed on the finger to measure PPG for SpO2, peripheral arterial tonometry, and distal skin temperature [14]. The two devices are time-synchronized and connect to a tablet via Bluetooth. Acceleration was recorded at 52 Hz for the *x*- and *y*-axes and 416 Hz for the *z*-axis (anteroposterior plane). ECG was recorded at 512 Hz, heart rate and respiratory rate at 1 Hz, PPG at 256 Hz, and skin temperature at 1 Hz. The system has been previously validated for sleep-related breathing disorders [15].

In addition, ActiWatch Spectrum (Philips, Cambridge, MA, USA) was used as a reference of a research-grade wearable sensor. ActiWatch was placed at the less affected side of the stroke patient. ActiWatch data were analyzed with the Autoscore sleep/wake algorithm from its software ActiWear (Philips, Cambridge, MA, USA).

### 2.3. Procedures

Initial screening was conducted via electronic medical records to identify patients who fulfilled the eligibility criteria, and medical clearance for participation was obtained from their main attending primary physician. Following consent, participants were asked to wear the PSG system and ANNE^TM^ sensors for a single night of their inpatient stay (Figure 1). Overnight recordings began at the participant’s normal bedtime in the hospital and ended either the next morning, when the participant awoke, or after 8 h of recording.

### 2.4. Data Analysis

All preprocessing was performed in MATLAB R2017b. Data visualization, segmentation, feature extraction, and machine learning were performed in Python 3.9.7 using the numpy, scikit-learn, imblearn, and pandas libraries [16]. Statistical analyses were performed in R 4.1.0 with the caret package [17].

Signals from the PSG system and ANNE sensors were time-synchronized via cross-correlation on their respective ECG signals (xcorr function in MATLAB). ANNE sensor data were cleaned by resampling to the expected sampling rate to ensure consistency, filtering, and extracting features for algorithm training and testing. A highpass fifth-order Butterworth filter was applied to accelerometer and ECG data with cutoff frequency at 1 Hz. PPG data were processed using the ANNE system’s proprietary software (Sibel Health, Inc., Niles, IL, USA) to obtain respiratory rate and oxygen saturation index (SpO2) at the frequency of 5 Hz. The Pan−Tomkins algorithm [18] was applied to the ECG time series signal to detect R peaks, which were used to compute R-R intervals and subsequent features related to heart rate and heart rate variability.

#### 2.4.1. PSG for Ground Truth Sleep Staging

PSG data were preprocessed using Brain Vision software (Morrisville, NC, USA). A Registered Polysomnographic Technologist visually scored each 30-s epoch of the PSG data as Wake, N1, N2, N3, or REM, in accordance with the American Academy of Sleep Medicine criteria [19]. These scores served as the ground truth for training and testing a machine learning algorithm to classify sleep stage based on the wearable sensor signals. To explore the various resolutions of sleep staging, we labeled the PSG scores for various sleep staging resolutions, including: 2-stage (Wake vs. Sleep (N1, N2, N3 and REM]), 3-stage (Wake vs. Non-REM Sleep (N1, N2, N3) vs. REM Sleep), and 4-stage (Wake vs. Light Sleep (N1 and N2) vs. Deep Sleep (N3) vs. REM Sleep).

#### 2.4.2. Feature Extraction

Sensor features were computed for each 30 s epoch, without overlap, and paired with the corresponding ground truth sleep stage from PSG scores. Features were generated for each sensor modality in the time and frequency domains, resulting in 73 total features (Table 2). Time domain features included the mean, standard deviation (STD), interquartile range (IQR), kurtosis, root mean square (RMS), variance, maximum, minimum, range, and inter-axis correlations. The following features related to heart rate were calculated: (1) number of successive R-R intervals that differ by more than 20 ms and 50 ms (i.e., NN20, NN50), and (2) percentage of successive R-R intervals that differ by more than 20 ms and 50 ms (i.e., PNN20, PNN50). A Fast Fourier transformation was used to estimate the power spectral density of the processed R-R intervals. These frequency domain features include zero crossing rate and power in the very low frequency (VLF), low frequency (LF), and high frequency (HF) bands. VLF was defined as the band from 0.0033 to 0.04 Hz, LF as 0.04 to 0.15 Hz, and HF as 0.15 to 0.4 Hz. Time spent in apnea (TSA) was calculated as the amount of time that SpO2 fell below certain thresholds, including 95%, 90%, 85%, 80%, and 70%. Oxygen desaturation index (ODI) was calculated as the number of times within each 30 s epoch that SpO2 decreased by a certain threshold from the previous epoch, including 2, 3, 4, and 5%. Features relating to the distal-to-proximal gradient (DPG) of skin temperature, which is the difference between the limb and chest temperature [20], were also calculated. Data were excluded if the minimum SpO2 was less than 50% or the minimum heart rate was less than 40, since these values are physiologically unrealistic for this patient cohort, and the features likely stemmed from noisy sensor signals.

#### 2.4.3. t-Distributed Stochastic Neighbor Embedding (tSNE) Analysis

We applied tSNE, a non-linear dimensionality reduction technique, to visualize our high-dimensional feature set in two-dimensional space. tSNE graphs illustrate the similarity of data points across multiple features using probability-based clustering [21]. Following parameter exploration, we set perplexity to 40 and iteration to 300 for tSNE clustering. All other parameters were set to the default values from Python scikit-learn (version 1.0.2).

#### 2.4.4. Class Imbalance

Class imbalance is an issue for most of the machine learning-based sleep monitoring system. Based on the four-sleep stage labels, we have collected 3484 epochs of light sleep (N1 and N2), 1237 epochs of REM sleep, 636 epochs of deep sleep (N3), and 469 epochs of wake. Since feeding a large amount of imbalanced training data into the model could result in overfitting toward the majority class, we remediate this situation with the following methods: (1) for random forest and bagging, we used class weights and random undersampling, and (2) for Gradient Boosting and XGBoost, we used the Synthetic Minority Oversampling Technique (SMOTE) [22].

#### 2.4.5. Model Development and Training

We constructed two types of supervised machine learning models, including population models (trained on data from a subset of subjects and tested on data from a held-out subject) and personal models (trained on a subset of data from one subject and tested on held-out data from that same subject). Four algorithms (Balanced Bagging, Balanced Random Forest, Gradient Boosting, and XGBoost) were compared in both the population and personal model frameworks for their ability to classify sleep stages at three different resolutions (2-, 3-, and 4-stage). In our initial exploration, we considered additional models, including linear discriminant classifiers, support vector machines, k-nearest neighbor, and a combination of recurrent and convolutional neural networks [23] for this problem. However, none of those models outperformed the ensemble and boosting algorithms presented here.

Prior to model training, correlated features (Pearson correlation coefficient greater than 0.95) were removed. All remaining features were normalized within subjects using z-score normalization to minimize the potential influence of baseline values and outliers on the model.

Models were trained and tested using a nested cross-validation approach with an inner and outer loop. For population models, the outer loop consisted of leave-one-subject-out cross-validation (LOSOCV), wherein each subject was considered a separate fold. For personal models, the outer loop consisted of 10-fold cross-validation, wherein the data were divided evenly into 10 folds and stratified by class so that each labeled sleep stage was equally represented across folds. For both types of models, cross-validation was implemented by assigning one fold as the test set and the remaining folds as the training set; this process was repeated until each fold was tested, and model performance metrics were averaged across all tested folds. The inner loop consisted of feature selection and hyperparameter tuning with an additional 10-fold cross validation. Using recursive feature elimination (RFE) and grid search, respectively, features and hyperparameters were selected based on maximization of the weighted F_1_ score.

#### 2.4.6. Model Evaluation

Sleep stage predictions from each model were compared to ground truth PSG labels for each test dataset. Cohen’s kappa was used for between-model comparisons and to quantify degree of agreement between the model predictions and ground truth. Cohen’s kappa was interpreted with values <0 as worse-than-chance agreement, 0–0.20 as slight, 0.21–0.40 as fair, 0.41–0.60 as moderate, 0.61–0.80 as substantial, and 0.81–1 as near-perfect agreement [24].

A pooled Cohen’s kappa was computed using all test data of each model to summarize the overall algorithm performance. We chose to pool rather than average the Cohen’s kappa values because averaging would have an additive effect on the numerator variance but a multiplicative effect on the denominator variance. This difference in variance calculation may increase the error of estimate especially around the middle values (e.g., Cohen’s kappa of 0.4–0.7) [25]. We also computed the following metrics for each class: (1) specificity, (2) precision, (3) sensitivity, (4) F_1_ score, and (5) balanced accuracy. Balanced accuracy is the arithmetic mean of recall for each class of the data. The F_1_ score is a harmonic average of precision and sensitivity, ranging from 0 (lowest performance) to 1 (highest performance), and is computed according to Equation (1):(1)$$F_{\beta }=\left(1+\beta^{2}\right)\cdot \frac{precision\cdot sensitivity}{\left(\beta^{2}\cdot precision\right)+sensitivity},\;\mathrm{where}\;\beta =1$$

## 3. Results

### 3.1. Sensor Data Visualization

After visual inspection, we excluded one subject from analysis due to consistently noisy ECG and PPG signals throughout the night, suggesting the sensors had poor contact with the skin during recording. Therefore, the final dataset for population models included nine subjects, amounting to 48.55 total hours of data for training and testing. After removing noisy data, Subject 6 had only 1.73 h available and was removed from the personal model analysis due to the small size of the dataset for training and testing a personal model for this participant. For personal models, a total of 46.82 h of data were available for use.

The ideal scenario for building a successful population model for sleep stage recognition is to have highly similar signal patterns and features within each sleep stage that generalize across individuals. For the subacute stroke population, we considered that each patient could have substantially different physiological signals (i.e., heart rate variability, oxygen saturation, and core-limb temperature fluctuations) due to the heterogeneous nature of stroke and its impact on regulation of the autonomic nervous system. Figure 2 illustrates this between-subject heterogeneity via tSNE graphs. Sleep stages are generally unable to explain different data clusters (Figure 2A), indicating a low probability that data points from the same sleep stage are similar. Rather, many of the tSNE clusters are comprised of data from individual subjects (Figure 2B). This suggests that, in the current dataset, similarity within subjects supersedes similarity within sleep stages, which would likely create challenges for a population-trained model to generalize to new patients.

### 3.2. Machine Learning

Cohen’s kappa scores for the different algorithms (Bagging, Random Forest, Gradient Boosting, and XGBoost) and sleep-stage resolutions (2-, 3-, and 4-stage) are given in Table 3 for both the population and personal model frameworks. As expected, population models generally performed poorly, with the highest Cohen’s kappa value of 0.27 provided by Gradient Boosting and 2-stage classification. Personalized models performed better by comparison, with the highest Cohen’s kappa value of 0.66 provided by XGBoost and 2-stage classification. The 2-stage classification from these models all outperformed the ActiWatch Autoscoring algorithm for these patients, with a Cohen’s kappa value of 0.48 (Table 3). Gradient boosting and XGBoost performed similarly across all metrics. Since both algorithms are methodologically similar, we selected XGBoost as the representative algorithm to evaluate performance in further analyses. Table A2 (Appendix A) shows detailed statistics of the population model using XGBoost algorithm.

Model performance metrics—including sensitivity, precision, F_1_-score, and balanced accuracy—for the best-performing model (XGBoost, personal) are given in Table 4 for each of the three levels of sleep-stage resolution. Macro average F_1_ scores were 0.83, 0.76, and 0.66 for the 2-, 3-, and 4-stage models, respectively. Figure 3 shows the confusion matrix of the 4-stage personal model. Average recall of wake, light sleep, deep sleep, and REM sleep were 75.3%, 65.0%, 83.3%, and 70.1%, respectively. Wake, deep sleep, and REM sleep were all most often misclassified as light sleep, while misclassifications of light sleep were similarly distributed across the other stages. Figure 4 shows a representative hypnogram from the best-performing personal model compared to ground-truth sleep stages from PSG. For this patient, the model slightly overestimated the time spent awake (4.9% of the total recording time, versus the actual 3.6% determined from PSG), underestimated light sleep (40.0%, versus actual 42.6%), overestimated deep sleep (32.0%, versus actual 27.8%), and underestimated REM sleep (23.2%, versus actual 26.0%). The model-estimated total sleep time was 357 min, versus 362 min from PSG. The model-estimated number of awakenings was six, versus one from PSG.

## 4. Discussion

In this preliminary study, we explored two types of models (population vs. personal), four algorithms (Balanced Bagging, Balanced Random Forest, Gradient Boosting, and XGBoost), and three resolutions of sleep staging (2-, 3-, and 4- stage). This is the first machine learning-based study to our knowledge which utilizes multimodal physiological data (motion, ECG, PPG, skin temperature) from commercialized wearable sensors to classify sleep stages in a subacute stroke population. This study using low-profile, multimodal wearable sensors provides a critical first step for improving the accuracy, resolution, and feasibility of longitudinal sleep monitoring after stroke. Measuring detailed changes in sleep architecture over time can better help us understand the complex relationship between sleep and stroke recovery. Detecting and intervening for patients with poor sleep quality may improve their rehabilitation outcomes or prevent future recurrence of stroke.

The wearable sensors used in this study offer a less obtrusive option for long-term sleep stage monitoring than PSG for patients with stroke across the inpatient or outpatient care settings. While current research-grade sleep monitors such as ActiWatch are also unobtrusive compared to standard PSG, their low monitoring resolution (2-stage detection of sleep and wake) and accuracy (mean balanced accuracy = 0.72) may not be sufficient to capture overall sleep quality for the subacute stroke population. Indeed, our personal machine learning models with multidimensional sensor data demonstrated improved accuracy in 2-stage monitoring over the ActiWatch for patients with stroke (mean balanced accuracy = 0.83), and this approach enhanced the ability to perform 3- and 4-stage detection with similar performance.

Personal models outperformed a population model, likely due to the small sample size and the heterogeneity in physiological signals between patients. For personal models, XGBoost was the best-performing algorithm with an F_1_ score of 0.76 when identifying wake, Non-REM, and REM sleep stages and an F_1_ score of 0.83 when identifying wake and sleep. A 4-stage personal model successfully recalled 65.0–83.3% of stages on average. Light sleep was the stage most prone to error, both in being misclassified as other stages or in having other stages misclassified as light sleep. This may be due to the greater prevalence of these samples in the dataset (class imbalance), as well as more nuanced physiological changes during the transitions between wake and deeper stages of sleep. Although the model-estimated total sleep duration and relative composition of sleep stages were generally similar to the ground truth obtained from PSG, misclassifications from the model can skew other metrics of sleep quality based on the estimated sleep architecture (e.g., number of awakenings). This advocates for additional study to improve the sleep-stage classifier.

For real-world implementation, personal models would require clinicians to record at least one night of PSG and wearable sensor data for each new patient. Sleep stages could be obtained from wearable sensors alone on the subsequent nights. Although most IRFs rarely conduct PSG-based sleep studies for the patients, unless ordered by the physician, the continued accumulation of evidence about the relationship between sleep and stroke may encourage PSG recordings to become a more common practice in the future. If so, PSG may be conducted early in the IRF program for patients with stroke, and less intrusive wearable devices could be used for the subsequent nights to facilitate continuous, long-term sleep monitoring. Personal models may be a reserve option for accurate, individualized monitoring due to the heterogeneity of the stroke symptoms that likely affect the physiological data of post-stroke individuals.

Ideally, one would want to create a wearable sensor-based sleep stage detection system without conducting a PSG sleep study on each new patient, since PSG equipment is expensive, uncomfortable for the wearer, and requires significant time and resources to collect and score data. Therefore, a population model is more favorable than a personal model for real-world clinical implementation. We found that the population model performed poorly in this study, which most likely can be explained by the low sample size for model training and the heterogeneity of sensor features between patients (Figure 2). The current study implemented an approach similar to our prior research in sleep classification with healthy adults [12]. In this study, a population model was more effective in classifying 2- and 3-stage sleep, detecting wake and sleep with a recall of 74.4% and 90.0%, respectively, and detecting non-REM and REM with recall of 73.3%, 59.0%, and 56.0%, respectively. The disparate performance of population models between our previous and current studies further indicates that sleep models for the stroke population may have different needs than those for healthy controls.

Other researchers with larger sample sizes have also demonstrate greater efficacy of population models for healthy individuals. For example, Zhang and colleagues [26] found an F_1_ score of 0.6 that constructed a population, multilayers deep learning model from 39 subjects using heart rate and motion data from wearable sensors. In a separate study, Zhang and colleagues [27] found that a 0.69 Cohen’s Kappa score constructed a population, bidirectional long-short term memory model which was trained from 417 subjects of public PSG database and tested on 32 subjects based on heart rate and respiratory rate. Beattie and colleagues [28] found a 0.5 Cohen’s Kappa score constructed a population, linear discriminant model with 60 subjects based on heart rate and motion. Together, this research suggests that transfer learning or a larger sample size in healthy individuals would improve the performance of a 4-stage population model. Additional details about wearable and non-wearable methods for non-invasive sleep monitoring can be found in a recent review [29].

However, presence of stroke likely complicates this task even more due to the complex effects of stroke on biophysical outcomes depending on the lesion site, severity, and treatment. For example, subcortical lesions have a more drastic effect on the central ANS than cortical lesions, which in turn would have a different impact on bodily processes such as heart rate, respiration, and temperature regulation [13]. Prior studies have shown features of heart rate variability are used as biomarkers in identifying stroke and predicting the condition and outcome of stroke [30,31]. Additionally, multiple co-morbidities combined with polypharmacy (Table 1 and Table A1) would further diversify the physiological measures of individuals undergoing recovery and treatment after stroke.

For data-driven modeling, the predictive power of a model can only extend as far as the existing data boundaries. Classifications outside of the training data boundary may reduce accuracy or generalizability of the machine-learned model. By increasing the heterogeneity and sample size of the training data, we expect the model boundaries would be expanded for more accurate classification. Previous large-scale database studies have seen moderate success in population models for automatic sleep stage detection using physiological signals such as ECG and/or PPG isolated from PSG data in targeted patient groups [32]. For example, Sridhar and colleagues leveraged ECG data from the Sleep Heart Health Study (SHHS; N = 6705) and Multi-Ethnic Study of Atherosclerosis (MESA; N = 1619) to build a deep neural network (DNN). They obtained 77% accuracy and a Cohen’s kappa coefficient of ~ 0.66 in a 4-stage resolution model (i.e., Wake, Light, Deep, and REM) [33]. Korkalainen and colleagues collected PSG data from 894 OSA patients and built a DNN consisting of a convolutional network and recurrent network. This model had an accuracy of 80.1% with Cohen’s kappa of 0.65 for predicting three sleep stages [23]. Thus, it is possible that with additional training data from patients with subacute stroke, automated sleep staging classification with wearable sensors will be possible. Future work will consider incorporating additional data from healthy individuals and applying transfer learning methods to improve performance for patients with subacute stroke. In absence of a larger database for multimodal wearable sensor data, we have demonstrated that personal models may be a feasible alternative for stroke patients requiring detailed, long-term sleep monitoring.

### Limitations

There are several limitations to this study, which should be considered and addressed for future work. First, a small sample size was used to train and test the machine learning models, likely contributing to the poor generalization of the population model. Based on the variable nature of stroke and treatments (e.g., medication) on patient biometrics, the optimal size and composition of training data for a generalized population model remains unknown. Future work may consider developing separate models for subsets of the stroke population, such as by stroke type or level of impairment, or incorporating data from healthy individuals to supplement the model training.

Second, all data were collected during a single night of recording. Multiple nights of collection will be a critical extension of this study to validate personal models (i.e., training an algorithm from one night of sensor data and testing on another). Although the 10-fold cross-validation method for personal models, necessitated by the single night of data, may potentially limit the implication of our results, it is an efficient way to pilot the feasibility of collecting wearable sensor data from a subacute stroke population and detecting sleep stage from a machine learning model.

Third, we excluded patients with clinically diagnosed sleep disorders. This exclusion was intended to minimize the potentially confounding impact of these disorders on the model in this preliminary study, since sleep disorders can additionally affect physiological signals beyond the already-variable effects of stroke (e.g., altered movement during REM sleep behavior disorders [34], altered patterns in heart rate variability, respiratory rate, and blood oxygenation for sleep apnea [35,36]). However, sleep disorders are common in the stroke population, with an estimated 50–70% of patients experiencing sleep apnea alone [37]. Thus, it will be essential to include patients with sleep disorders in future model training and validation efforts.

## 5. Conclusions

In this study, we established feasibility to construct machine-learning models for sleep monitoring in a subacute stroke population using data from two wireless, wearable sensors. We examined both population and personal models for the supervised classification of sleep stages. The heterogeneity of biophysical signals after stroke will pose a challenge in building a population model that generalizes across individuals, likely requiring significantly more training data. Personal models were a feasible alternative for a small sample size, demonstrating fair accuracy in distinguishing wake, non-REM, and REM sleep. Future work will sample additional data across and between patients with subacute stroke for robust model training and refinement.

## Figures and Tables

**Figure 1 sensors-22-06190-f001:**
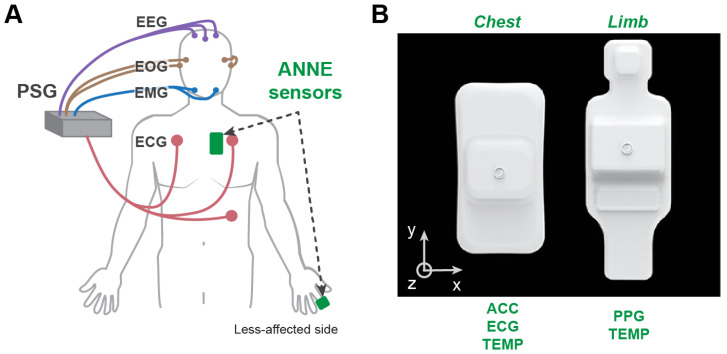
**System configuration for development of a sleep monitoring algorithm in patients with stroke.** (**A**) Placement of PSG electrodes, ActiWatch^TM^, and ANNE^TM^ One wearable sensors. ANNE^TM^ system included two wireless devices that were placed on the chest (left of midline) and the limb (index finger of the less-affected side). (**B**) The ANNE^TM^ devices record multimodal physiological data and are encapsulated with soft, flexible materials. EEG = electroencephalography; EOG = electrooculography; EMG = electromyography; ECG = electrocardiography; PPG = photoplethysmography; ACC = tri-axial acceleration; TEMP = skin temperature.

**Figure 2 sensors-22-06190-f002:**
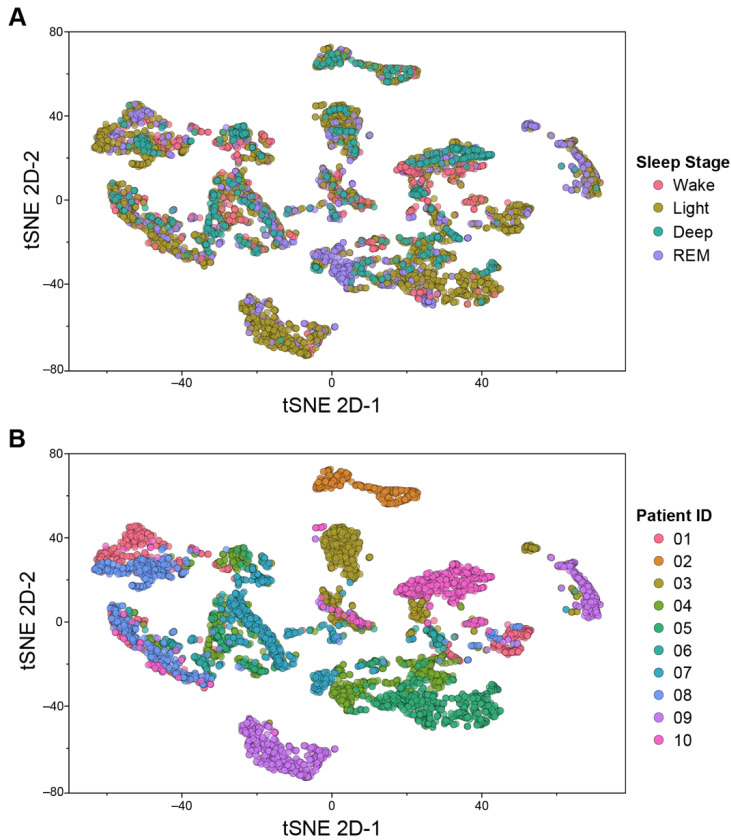
**tSNE plot of sensor features from 10 patients with stroke.** Representation of similarities among sensor features in two-dimensional space, color-coded by (**A**) sleep stage and (**B**) patient. The clusters illustrate that features are more similar within patients than within sleep classes, suggesting that the similarity within each subject is greater than similarities across different sleep stages. This indicates a machine learning algorithm trained on population data may be challenged to learn characteristic patterns of the different sleep stages that would generalize to new patients.

**Figure 3 sensors-22-06190-f003:**
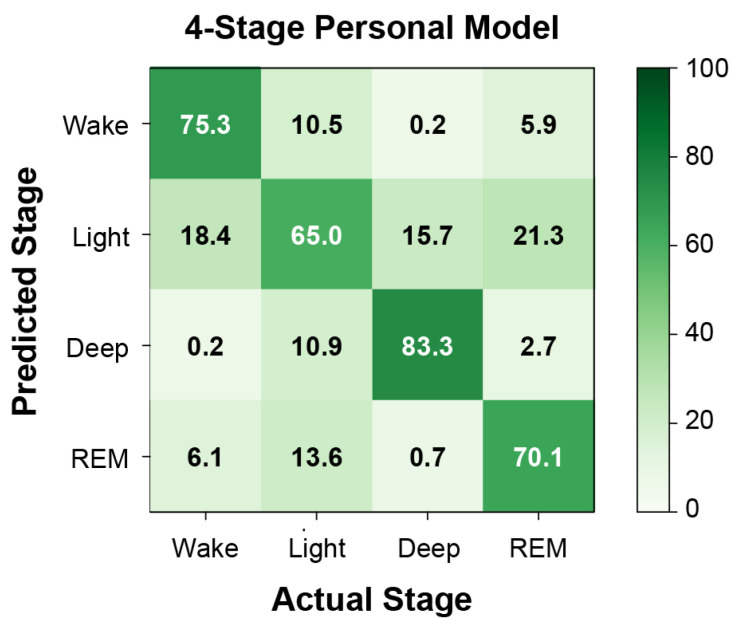
**Confusion matrix for 4-stage personal model.** Percentage of epochs correctly and incorrectly predicted from wearable sensor data, separated by actual sleep stage. Percentages are calculated from the total number of predicted and actual epochs, aggregated across all participants.

**Figure 4 sensors-22-06190-f004:**
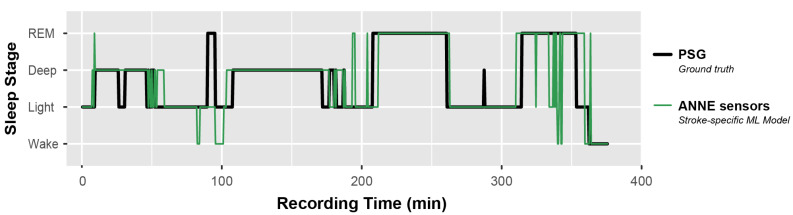
**Example hypnogram for 4-stage personal model.** Comparison of the predicted sleep stage from wearable sensor data (green line) and the actual sleep stage from PSG (black line) throughout the night for a single participant (ID 4).

**Table 1 sensors-22-06190-t001:** Demographics and clinical characteristics of study participants.

ID	Age	Sex	BMI	Race	Stroke Type	Affected Side (Left/Right)	Experiencing Pain (Yes/No, Self-Report)	No. of Medications with Sleep-Related Side Effects (Drowsiness, Insomnia)
1	53	M	23.6	C	Isc+Hem	L	Y	2 D, 2 I
2	52	F	40.3	AA	Isc	R	N	0 D, 3 I
3	56	M	38.6	C	Isc	R	N	2 D, 2 I
4	48	F	30.4	C	Isc	L	N	1 D, 1 I
5	64	F	27.5	AA	Hem	L	Y	1 D, 4 I
6	70	F	21.0	AA	Isc	L	N	1 D, 5 I
7	37	F	32.6	PI	Isc	R	Y	2 D, 3 I
8	56	M	39.1	NA	Isc	L	Y	2 D, 2 I
9	65	M	25.8	A	Hem	L	N	0 D, 1 I
10	80	M	23.1	C	Isc	L	N	0 D, 2 I
Mean (SD)or Count	58.1 (12.1)	5 F, 5 M	30.2 (7.2)	4 C, 3 AA, 1 A, 1 PI, 1 NA	7 Isc, 2 Hem, 1 Isc+Hem	7 L, 3 R	4 Y, 6 N	1.1 D, 2.5 I

Isc = Ischemic; Hem = Hemorrhagic; C = Caucasian; AA = African American; PI = Pacific Islander; NA = Native American; A = Asian; D = Drowsiness; I = Insomnia.

**Table 2 sensors-22-06190-t002:** Features extracted from ANNE^TM^ sensor data during overnight monitoring.

Sensor Modality	Sampling Freq (Hz)	No. of Features	Features		
ACC (Chest)	52	33	Mean (x, y, z) Min (x, y, z) Max (x, y, z) Range (x, y, z)	IQR (x, y, z) SD (x, y, z) Kurtosis (x, y, z) RMS (x, y, z)	Variance (x, y, z) rho (x, y, z) p (x, y, z)
ECG	512	19	HR mean HR min HR max SDNN RMSSD NN50 NN20	PNN50 PNN20 VLF power VLF peak LF power LF peak	HF power HF peak LFHF ratio R-R mean R-R min R-R max
TEMP	5	6	DPG mean DPG min	DPG max DPG range	Chest (proximal) mean Limb (distal) mean
PPG	256	15	SpO2 mean SpO2 min SpO2 variance SpO2 rho SpO2 ZC	SpO2 DI TSA95 TSA90 TSA85 TSA80	TSA70 ODI2 ODI3 ODI4 ODI5

rho = correlation coefficient; p = correlation p-value; IQR = interquartile range; SD = standard deviation; RMS = root mean square; HR = heart rate; NN*x* (or PNN*x*) = sum (or percentage) of R-R intervals larger than *x* ms (or %); LF = low frequency; VLF = very low frequency; HF = high frequency; DPG = distal-to-proximal gradient; TSA*x* = time spent in apnea, with SpO2 below *x*%; SpO2 DI = mean absolute difference between successive mean values of SpO2 over 10-s intervals; ODI*x* = oxygen desaturation index for SpO2 dropping *x*% from the previous epoch; ZC = zero crossing rate.

**Table 3 sensors-22-06190-t003:** Models and ActiWatch Autoscore comparisons with pooled Cohen’s kappa.

Algorithm	Sleep Stage Resolution (No. Classes)	Population Model	Personalized Model
Bagging Classifier	2	0.249	0.483
	3	0.132	0.473
	4	0.003	0.527
Random Forest	2	0.248	0.577
	3	*0.171*	0.532
	4	*0.061*	0.517
Gradient Boosting	2	*0.268*	0.549
	3	0.110	0.602
	4	0.037	*0.617*
XGBoost *	2	0.249	*0.660*
	3	0.128	*0.600*
	4	0.014	0.531
ActiWatch Autoscore	2	0.477	

Italic values indicate the best-performing algorithm within each sleep stage resolution (2-stage, 3-stage, 4-stage) for both population and personalized models. Asterisk (*) indicates the algorithm selected for further analysis, based on its best or near-best performance across model designs.

**Table 4 sensors-22-06190-t004:** Comparison of ActiWatch Autoscore and XGBoost algorithm performance (mean and SEM) for personal models, including 2-stage, 3-stage, and 4-stage sleep detection.

Sleep Stage	Specificity	Precision	Sensitivity	F_1_	Balanced Accuracy
2-stage					
Wake	0.97 (0.01)	0.81 (0.05)	0.68 (0.06)	0.68 (0.04)	0.83 (0.03)
Sleep	0.68 (0.06)	0.96 (0.01)	0.97 (0.01)	0.97 (0.01)	0.83 (0.03)
3-stage					
Wake	0.94 (0.01)	0.78 (0.03)	0.81 (0.04)	0.74 (0.03)	0.88 (0.02)
NREM	0.80 (0.03)	0.90 (0.02)	0.81 (0.03)	0.83 (0.04)	0.81 (0.03)
REM	0.89 (0.03)	0.75 (0.05)	0.76 (0.04)	0.71 (0.05)	0.82 (0.03)
4-stage					
Wake	0.96 (0.01)	0.78 (0.04)	0.77 (0.04)	0.72 (0.03)	0.87 (0.02)
Light	0.79 (0.03)	0.73 (0.09)	0.63 (0.07)	0.66 (0.08)	0.71 (0.05)
Deep	0.91 (0.02)	0.57 (0.10)	0.70 (0.11)	0.58 (0.10)	0.76 (0.10)
REM	0.86 (0.04)	0.69 (0.05)	0.74 (0.05)	0.67 (0.05)	0.80 (0.04)
ActiWatch Autoscore Algorithm (2-stage)					
Wake	0.92 (0.02)	0.56 (0.05)	0.51 (0.08)	0.50 (0.07)	0.72 (0.04)
Sleep	0.52 (0.09)	0.90 (0.02)	0.93 (0.02)	0.91 (0.02)	0.72 (0.04)

## Data Availability

The dataset may be made available to an investigator upon request for academic, research, and non-commercial use, subject to any license.

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
