# Peer review of "Sleep Monitoring during Acute Stroke Rehabilitation: Toward Automated Measurement Using Multimodal Wireless Sensors"

_sensors, 2022, doi:10.3390/s22166190_

Round 1

Reviewer 1 Report

This manuscript presents a wearable sensor-based approach using machine learning to monitor the sleep stages in stroke recovery. The idea of the paper is interesting and can help in the cost-effective and non-intrusive monitoring of sleep in recovering patients. The paper is well-written and the overall structure is easy to follow. I have the following comments/suggestions for the authors to further improve the quality of the manuscript.

1. I have a major concern about the sample size used for the validation of the proposed approach. The authors have used just one night of data from nine subjects (even less for the personal model). It is quite possible that subjects will have variation in sleep on different nights. It is very important that the proposed model is able to handle these variations.

2. Also, they have excluded the subjects with sleep disorders (such as apnea) but in reality, many people especially in old age, have at least one of these sleep disorders. It would be interesting to include subjects with sleep disorders.

3. The population model is the most suited model to solve the problem of non-intrusive sleep monitoring but the proposed model has shown poor performance for the population model. Given the limited dataset, I am not sure whether the proposed model would be able to predict the sleep stages with reasonable accuracy if more subjects are included in the study (which reflects the reality).

4. My other concern is about the applicability of the personal model. Is it practical to train the model for each new patient (use PSG as ground truth) and then monitor the sleep?

5. Although, cross-validation is a good technique to train the model in the case of limited data, it can limit the generalization of the model and the model performance will not be good when exposed to new subjects. The best option, in this case, should be the LeaveOneSubjectOut or LeaveOneGroupOut (if multiple nights are recorded for each subject) which will generalize the model.

6. It would also be good to compare this work with some recent works which use a similar approach i.e., wearable-based sleep monitoring. The following survey summarizes some of the recent works for sleep monitoring including the wearable-based. The authors may use some of the listed techniques for comparing with their proposed model.

Hussain, Z., Sheng, Q.Z., Zhang, W.E., Ortiz, J. and Pouriyeh, S., 2022. Non-invasive techniques for monitoring different aspects of sleep: a comprehensive review. ACM Transactions on Computing for Healthcare (HEALTH)3(2), pp.1-26.  

Reviewer 2 Report

In the manuscript of sensors-1839907, the authors collected overnight biophysical data from patients with subacute stroke using a simple sensor system and constructed machine-learned algorithms to detect sleep stages. The reseach is well-organized and offer a provisional method to capture high-resolution sleep metrics from simple wearable sensors by leveraging a single night of polysomnography data. The following comments are provided for the authors.

1. In the section of introduction, the importance of sleep monitoring for acute stroke rehabilitation should be discussed.
2. Figure 3 shows the confusion matrix of the 4-stage model, and the authors should discuss the figure in the text.
3. Figure 4 shows a representative hypnogram from the best-performing personal model compared to ground-truth sleep stages from PSG. The authors should give the detailed analysis of similarities and differences betweenthe the predicted sleep stage from wearable sensor data and the actual sleep stage from PSG, and what the new approach might mean.

Round 2

Reviewer 1 Report

The authors have addressed all of my comments and I am satisfied with the response.

Reviewer 2 Report

All the comments and suggestions are addressed, and I recommend the manuscript to be accepted in present form .